# Adsorption Thermodynamics and Dynamics of Three Typical Dyes onto Bio-adsorbent Spent Substrate of *Pleurotus eryngii*

**DOI:** 10.3390/ijerph16050679

**Published:** 2019-02-26

**Authors:** Jianguo Wu, Aiqiang Xia, Chunyan Chen, Liuying Feng, Xiaohui Su, Xinfeng Wang

**Affiliations:** 1Jiangsu Key Laboratory for Eco-Agricultural Biotechnology around Hongze Lake, Jiangsu Key Construction Laboratory for Food Safe and Nutritional Function, School of Life Science, Huaiyin Normal University, Changjiang West Road 111, Huai’an 223300, China; aiqingxia163@163.com (A.X.); chunyanchen0905@163.com (C.C.); 18360908332@163.com (L.F.); 15751333929@163.com (X.S.); wangxf@hytc.edu.cn (X.W.); 2Jiangsu Collaborative Innovation Center of Regional Modern Agriculture and Environmental Protection, Huaiyin Normal University, Changjiang West Road 111, Huai’an 223300, China

**Keywords:** bio-adsorption, thermodynamics, isotherms, kinetics, spent mushroom substrate

## Abstract

Dyeing wastewater is very hard to treat, and adsorption could be a good choice. Spent substrate of *Pleurotus eryngii* (SSPE) was first used to adsorb malachite green, safranine T and methylene blue from aqueous solutions, and the corresponding adsorption isotherm, thermodynamics and dynamics models were simulated. More than 93% of the dyes were removed with solutions with 100 mg/L of initial dye concentration, 1 g of SSPE and pH of 6.0 after adsorption for 4 h. Freundlich isotherm models fit better the adsorption data than Langmuir models. Adsorption of the dyes onto SSPE was a spontaneous exothermic process based on an adsorption thermodynamics model. SSPE could adsorb the dyes rapidly, and a second-order kinetics model fit better with the adsorption data than a pseudo first-order kinetics model. Accordingly, SSPE could be a good bio-adsorbent for the removal of malachite green, safranine T and methylene blue from the aqueous solution.

## 1. Introduction

Dyeing wastewater is discharged from cotton, linen, chemical fibers and their blended products, silk-based printing and dyeing, wool dyeing and finishing and silk factories. Dyeing wastewater is very hard to treat because of its large amount of water, high content of organic pollutants, high alkalinity and great change of water quality, which results in dyeing wastewater pollution [1]. Even more serious, many dyes threaten human health seriously because of their toxic, carcinogenic and mutagenic effects [2,3,4,5]. Dyes in the dyeing wastewater could be reused by adsorption and desorption [6]. In recent years, some new synthetic adsorbents and bio-adsorbents originated from agriculture and food industry waste for the removal of dyes wastewater was found and have attracted a lot of attention, such as Amberlite IRA-938 resin [7], starch/poly(alginic acid-*cl*-acrylamide) nanohydrogel [8], orange peel, sawdust, rice husk and spent substrate of *Ganodorma lucidum* [9,10,11,12]. Basic dyes are one of the major categories of dyes, and are suitable for dyeing of acrylic fibers, terylene, nylon, cellulose and protein fiber. Typical basic dyes include the triarylmethane dye malachite green, the azine dye safranine T and the thiazine dye methylene blue. Their chemical structures were shown in Figure 1.

In 2011, the mushroom yield in China was over 20 million tons, which was seventy five percent of the world output [13]. The turnout of spent mushroom substrate (SMS) was more than 8 million tons. SMS had thus become one of the major agriculture wastes. Except for a few examples of SMS being used for returning to the field [14], feedstuffs [15], cultivation [16], fuel and energy materials [17], the vast majority of SMS was abandoned and caused environmental pollution. Because of the abundant surface hydroxyl, carbonyl, and carboxyl groups in SMS [12], SMS could be considered as a potentially useful low-cost bio-adsorbent for dyeing wastewater treatment. However, there was only a few studies using SMS as bio-adsorbent to adsorb dyes and heavy metals [18,19,20].

*Pleurotus eryngii* has been widely cultivated all over the world, and its automated batch production has been realized. However, the utilization of spent substrate of *P. eryngii* (SSPE) is still a difficult problem [21]. In this study, SSPE was used for the first time as a bio-adsorbent for the adsorption of three typical dyes: the triarylmethane dye malachite green, the azine dye safranine T and the thiazine dye methylene blue from aqueous solutions, and the corresponding adsorption isotherm, thermodynamics and dynamics models were also simulated.

## 2. Materials and Methods 

### 2.1. Preparation of Bio-Adsorbent

SSPE was obtained from an edible fungus factory located in Huai’an, China, and washed with tap water first, then distilled water. The SSPE was milled and sieved after it was dried at 60 °C for 72 h. The particles ≤ 0.25 mm was picked for use as bio-adsorbent. 

### 2.2. Preparation of Dye Solution

Three stock solutions of 200 mg/L were prepared by dissolving 100.0 mg of malachite green (Shenyang Chascitek Chemical Co., Ltd, Shenyang, China), safranine T (Shanghai Fortunei Bio-tech Co., Ltd, Shanghai, China) or methylene blue (Shanghai Yuanye Bio-tech Co., Ltd, Shanghai, China) in 500 mL of double-distilled water. All the chemicals were analytical-grade reagents. The initial pH was adjusted with 0.1 mol/L HCl or 0.1 mol/L NaOH.

### 2.3. Batch Bio-Adsorption Procedure

Our batch bio-adsorption procedures included an experimental group and a control group. The batch bio-adsorption of experimental group was conducted in 250 mL conical flasks with 100 mL solution of malachite green, safranine T or methylene blue (10, 20, 50, 80, 100 and 120 mg/L of concentration) mixed individually with 1 g of SSPE under different conditions of pH (2.0, 4.0, 6.0, 8.0, 10.0 and 12.0) and absolute temperature (288, 293, 298, 303, 308, 313, 318, 328, and 338 K) for different times (1, 3, 5, 7, 10, 20, and 30 min ) or with 4 h equilibrium time. All the adsorption experiments were carried out in the shaker controlled by a thermostat (uncertainty of ± 0.5 K). Control group experiments were also conducted under the same batch bio-adsorption procedure conditions of the experimental groups but without SSPE. 

### 2.4. Determination of Concentration of Dyes

The equilibrium suspensions were centrifuged at a speed (7000 r/min) for 10 min, and the concentrations of dyes were determined by a UV–vis spectrophotometric method using a UV–vis spectrophotometer (754, Shanghai Sunny Hengping Instrument Co., Ltd., Shanghai, China) with glass cells of path length 1 cm at wavelength *λ*_malachite green_ 617 nm, *λ*_safranine T_ 530 nm and *λ*_methylene blue_ 664 nm. All data were measured five times. 

### 2.5. Removal Efficiency by Adsorption and Equilibrium Adsorption Quantity

Removal efficiency by adsorption (R) of dye onto SSPE was calculated by Equation (1), and the equilibrium adsorption quantity of SSPE for dye, *Qe* (mg/g), was calculated by Equation (2).
(1)$$R=\frac{C_{0}-Ce}{C_{0}}\times 100\%$$
(2)$$Qe=(C_{0}-Ce)\times \frac{V}{W}$$
where *C*_0_ and *Ce* are the equilibrium liquid-phase concentrations of the dyes of the control group and experimental group, respectively (mg/L), *V* is the volume of the experimental solution (=0.1 L), and *W* is the weight of the SSPE used (=1 g).

### 2.6. Assays by Fourier Transform Infrared Spectroscopy (FTIR)

KBr pellets were prepared with 2 mg SSPE in 40 mg KBr for FTIR spectroscopy using a Nicolet iS50 spectrometer (Thermo Scientific, Waltham, MA, USA). Spectral measurement ranged from 400 to 4000 cm^−1^ with 0.4 cm^−1^ resolution.

### 2.7. Adsorption Thermodynamics and Dynamics Model

#### 2.7.1. Adsorption Isotherm Model

To evaluate the adsorption isotherm of dyes onto adsorbent, the Langmuir and Freundlich models are widely used [22]. The Langmuir model is suitable for single molecule adsorption, and can be expressed linearly by Equation (3) as follows:(3)$$\frac{Ce}{Qe}=\frac{Ce}{Qm}+\frac{1}{K_{L}\times Qm}$$

The Freundlich model provides an empirical description of the single component adsorption equilibrium, and is applicable within a broader scope. The Freundlich model can be expressed as Equation (4):(4)$$\ln Qe=\ln K_{F}+\frac{1}{n}\ln Ce$$
where *Ce* is the equilibrium liquid-phase concentrations of the dye (mg/L), *Qe* and *Qm* are the equilibrium and maximum adsorbed amount of dye per mass of SSPE, respectively (mg/g), *K_L_* is the Langmuir equilibrium adsorption constant (L/mg) related to the free energy of adsorption, *K**_F_* is the Freundlich constant [(mg/g) (L/mg)^1/*n*^] related to the strength of the adsorptive bond, and 1/*n* is the adsorption intensity factor or surface heterogeneity.

#### 2.7.2. Adsorption Thermodynamics

To evaluate the adsorption thermodynamics of dyes onto the adsorbent, Equation (5) is widely used [12]:(5)$$\ln\frac{C_{Be}}{C_{Ae}}=-\frac{\Delta H}{RT}+\frac{\Delta S}{R}$$
where *C_Be_* and *C_Ae_* represent the equilibrium concentrations of the dye in adsorbent and solution (mg/L), Δ*H* is adsorption enthalpy change (kJ mol^−1^), Δ*S* is adsorption entropy change (J mol^−1^ K^−1^), *R* is the molar gas constant (8.314 J mol^−1^ K^−1^), and *T* is the absolute temperature (K).

#### 2.7.3. Adsorption Kinetics

To evaluate the adsorption kinetics, the pseudo first-order (Equation (6)) and second-order kinetics (Equation (7)) shown below are used widely [23]:(6)$$\ln(Qe-Qt)=\ln Qe-k_{1}t$$
(7)$$\frac{t}{Qt}=\frac{t}{Qe}+\frac{1}{k_{2}Qe^{2}}$$
where *Qe* and *Qt* represent the adsorption capacities (mg/g) of SSPE at equilibrium and at a particular time *t* (min), respectively. The first-order and second-order kinetic rate constants are denoted by *k*_1_ (1/min) and *k*_2_ [(g/mg) (1/min)].

## 3. Results

### 3.1. Effect of pH on Dyes Adsorption

The effect of pH on the removal of malachite green, safranine T and methylene blue from the solutions is shown in Figure 2. 

When the initial pH of the dyes solutions was increased from 2 to 12, the adsorption removal efficiency of malachite green and methylene blue onto SSPE remained stable at about 95 %, so pH had little effect on the adsorption of malachite green and methylene blue. As the pH increased from 2 to 6, the adsorption removal efficiency of safranine T onto SSPE increased from 84 to 93 %. By further increasing the pH to 8, the adsorption removal efficiency changed little, while with an increase of the pH from 8 to 12, the adsorption removal efficiency of safranine T decreased from 93 to 88 %. Therefore, pH affected the adsorption of safranine T onto SSPE, and around neutral pH (6–8) was propitious for safranine T to be adsorbed onto the SSPE in solution. Safranine T (SH^+^) undergoes triplet excited state proton-transfer reactions, and SH^+^ reacted with protons or hydroxy ions to form the dication SH_2_^2+^ or the neutral ^3^S species. SSPE could have different adsorption capacity for safranine T in different states [24], which lead to the effect of pH on the removal of safranine T by SSPE.

SSPE is rich in functionalg groups such as -OH (3420 cm^−1^) and -COOH (1650 cm^−1^) according to the FTIR spectruma shown in Figure 3, which means that the SSPE surface is rich in anions. Malachite green, safranine T and methylene blue are all basic dyes, which are in a cationic state in water, so the dyes could be adsorbed onto SSPE by electrostatic attractions. SSPE was also used to adsorb acid the dye amaranth in this study, but it was found it was very hard to adsorb amaranth on SSPE. Consequently, SSPE should be a good bio-adsorbent for the basic dyes malachite green, safranine T and methylene blue, with adsorption removal efficiencies of more than 93%. The dyes on SSPE could be further recycled by extraction technology, and the adsorbent could be reused further by bio-pretreatment with white-rot fungi [12].

### 3.2. Adsorption Isotherm

The adsorption removal efficiency of malachite green, safranine T and methylene blue onto SSPE generally decreased with the increase of initial dye concentration from 10 mg/L to 120 mg/L, as shown in Figure 4a. 

To effectively evaluate the Langmuir isotherm, the values of *Ce*/*Q**e* were plotted vs *C**e* as shown in Figure 4b, and the values of *K**_L_* and *Q**m* were obtained as shown in Table 1 by the slope and intercept of the best fit line, respectively. Furthermore, the Freundlich isotherm parameters of 1/*n* and *K**_F_* were obtained by plotting the values of ln*Qe* vs ln*C**e* as shown in Figure 4c and Table 2. The regression coefficient (*R*^2^) was used to evaluate the fit effect of the Langmuir and Freundlich isotherms to the experimental data. *Qm* of safranine T was higher than that of malachite green and methylene blue. However, the *R*^2^ of safranine T was below 0.9, which meant that adsorption isotherm parameters of safranine T onto SSPE had low feasibility according to the Langmuir model. *K_F_* is related to the strength of the adsorptive bond. The value of *K_F_* of safranine T was lower than that of other dyes, which meant that SSPE had a weak adsorptive bond strength for safranine T. The value of *K_F_* of methylene blue was the highest, which meant that methylene blue was most effectively removed by SSPE. That result was also consistent with the adsorption data. However, the values of *K_F_* of malachite green, safranine T or methylene blue onto SSPE were all lower than that onto synthetic adsorbents, such as Jalshakti^®^ (Indian Organic Chemicals Ltd., Khopoli, Raigad, India) [25] or Fe_3_O_4_@AMCA-MIL-53(Al) nanocomposite synthesized by Alqadami etal [26]. As bio-adsorbent, the values of *K_F_* of methylene blue onto SSPE was lower than those of activated lignin-chitosan extruded blends [27] but higher than brown macroalgae [28]. Based on the alignment of the experimental data with the model lines in Figure 4b,c, together with the regression coefficient (*R*^2^) in Table 1 and Table 2, it was clear that Freundlich model fitted the adsorption data better than the Langmuir model, which means that the adsorption of dyes took place on the monolayer surface of SSPE and the dye and SSPE interacted with each other during the adsorption process. Values of “*n*” obtained from the Freundlich isotherm were all above 1, which indicates potential good adsorption of the dyes onto SSPE.

### 3.3. Adsorption Thermodynamics

Adsorption removal efficiency of malachite green, safranine T and methylene blue onto SSPE decreased with increase of adsorption temperature as shown in Figure 5a, which indicates that a higher temperature was unfavourable to the adsorption of the dyes onto SSPE.

The adsorption thermodynamics model was established by plotting the value of ln(*C_Be_*/*C_A_e*) against the reciprocal of the temperature (1/*T*) to calculate the values of Δ*H* and Δ*S* and the regression coefficients (*R*^2^) as shown in Figure 5b and Table 3. *R*^2^ of the adsorption thermodynamics model of the dyes onto SSPE were all greater than 0.98, which indicated that the model fit well with the data of the dyes adsorption onto SSPE. Δ*H* and Δ*S* were all negative. Negative enthalpy indicates that the adsorption of dyes onto SSPE was an exothermic process. Negative entropy means that the degree of system chaos decreased because the dissolved dyes were adsorbed onto SSPE.

Gibbs free energy (Δ*G*) was used as a criterion for spontaneity and equilibrium of constant temperature and pressure processes, and was defined as Equation (8) [29]:(8)ΔG=ΔH−ΔS×T

The Δ*G* of malachite green and methylene blue adsorbing onto SSPE were all negative, which indicated that the adsorption process was a spontaneous process [30]. Δ*G* of safranine T adsorbing onto SSPE at the temperature ranged from 288 to 308 K was negative also, but the Δ*G* was positive when the adsorption temperature was above 308 K, so the adsorption process of safranine T onto SSPE was a spontaneous process below 308 K, and a non-spontaneous process above 308 K. 

### 3.4. Adsorption Kinetics

The adsorption removal efficiency of the three dyes malachite green, safranine T and methylene blue onto SSPE increased with the increase of adsorption time as shown in Figure 6a. The adsorption removal efficiency of the three dyes all exceeded 86 % during the initial three minutes, and then increased slowly during the subsequent adsorption time. Therefore, SSPE could efficiently adsorb malachite green, safranine T and methylene blue. 

The pseudo first order kinetic model was established by plotting the value of ln (*Qe*–*Qt*) vs time (*t*) to calculate the values of *k*_1_ and *R*^2^ as shown in Figure 6b and Table 4. Moreover, the values of *k*_2_, *Qe*, and *R*^2^ of the pseudo second order kinetic model were obtained by plotting the values of *t*/*Qt* vs time (*t*) as shown in Figure 6c and Table 4. *R*^2^ values of the two kinetic models were compared to obtain which model fitted better with the dye adsorption onto SSPE data. The linear plot of t/*Q**t* vs *t* as shown in Figure 6c indicates the applicability of the second-order kinetic model and good agreement with the obtained experimental data. The adsorption capacities of malachite green, safranine T and methylene blue onto SSPE at equilibrium (*Qe*) were 9.662, 9.389 and 9.823 mg/g, respectively. The *R*^2^ of the second-order kinetic model was much better than that of the first order kinetic model, which indicated that second-order kinetic model was suitable to describe the adsorption kinetics of the dyes. 

In a word, the results obtained in this study suggest that SSPE has the potential to be an economic and efficient bio-adsorbent for malachite green, safranine T and methylene blue removal from water and wastewater resources. A filtration unit incorporating SSPE as bio-adsorbent could be operated based on the conditions, adsorption thermodynamics and dynamics model obtained in this study in a full-scale water and wastewater treatment plant for the removal of malachite green, safranine T and methylene blue from contaminated waters.

## 4. Conclusions

SSPE should be a good bio-adsorbent for the basic dyes malachite green, safranine T and methylene blue, as the adsorption removal efficiencies were more than 93%. Freundlich isotherm models fit better with the adsorption data than Langmuir models. Adsorption of the dyes onto SSPE was a spontaneous exothermic process based on adsorption thermodynamics model. SSPE could adsorb the three dyes rapidly, and kinetic analysis of the dyes adsorption onto SSPE showed a good alignment with the second-order kinetics model.

## Figures and Tables

**Figure 1 ijerph-16-00679-f001:**
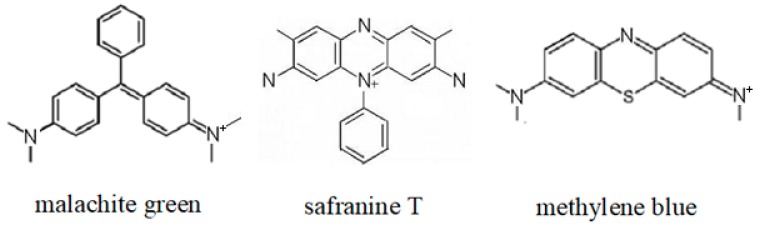
The chemical structures of malachite green, safranine T and methylene blue.

**Figure 2 ijerph-16-00679-f002:**
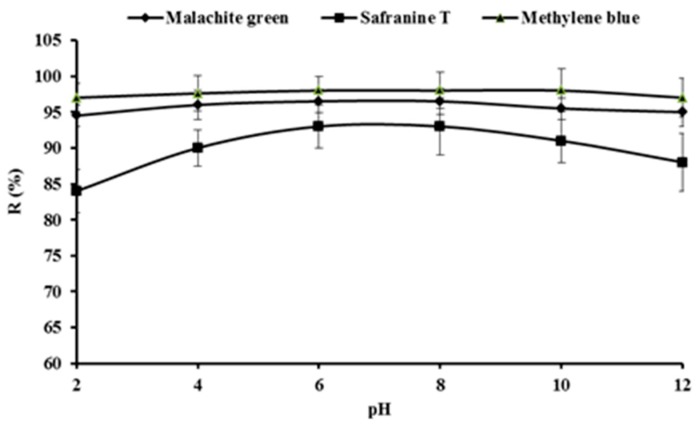
Adsorption removal efficiency of malachite green, safranine T and methylene blue onto SSPE under different pH in solutions at 100 mg/L of initial dyes concentration, 1g of SSPE and 303 K of temperature for 4 h.

**Figure 3 ijerph-16-00679-f003:**
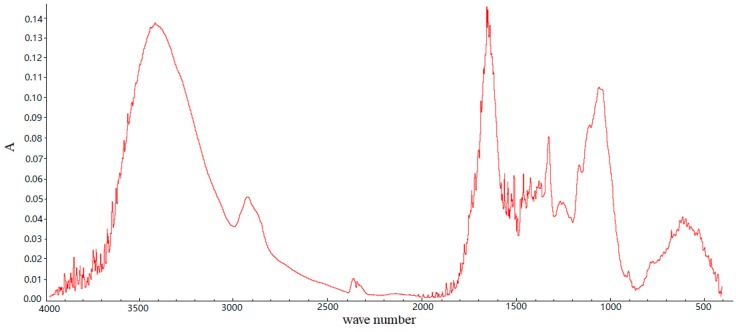
The FTIR spectrum of SSPE.

**Figure 4 ijerph-16-00679-f004:**
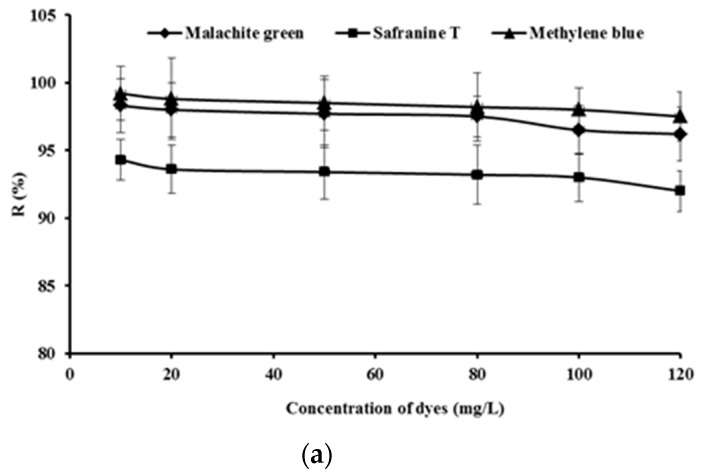
Adsorption removal efficiency of malachite green, safranine T and methylene blue onto 1 g of SSPE with different initial dye concentrations, at pH 6.0 and 303 K of temperature for 4 h (**a**), plots of Langmuir (**b**) and Freundlich (**c**) isotherms. The solid lines are the experimental data, and the dashed lines are the linearized Langmuir and Freundlich fitted models.

**Figure 5 ijerph-16-00679-f005:**
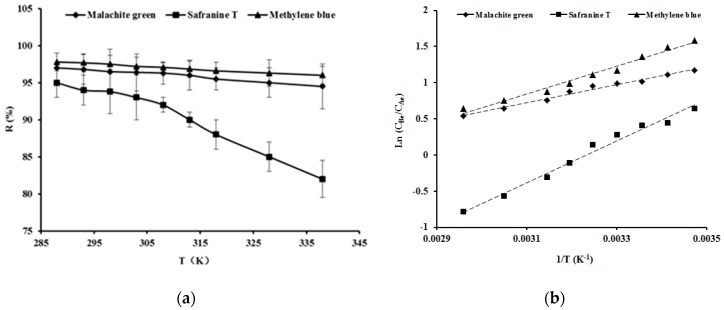
Adsorption removal efficiency of malachite green, safranine T and methylene blue onto SSPE with different temperature at 100 mg/L of initial dyes concentration, 1g of SSFV and 6.0 of pH for 4 h (**a**), and plots of adsorption thermodynamics (**b**). The solid lines were experimental data, and dashed lines were the linearized adsorption thermodynamics fitted model.

**Figure 6 ijerph-16-00679-f006:**
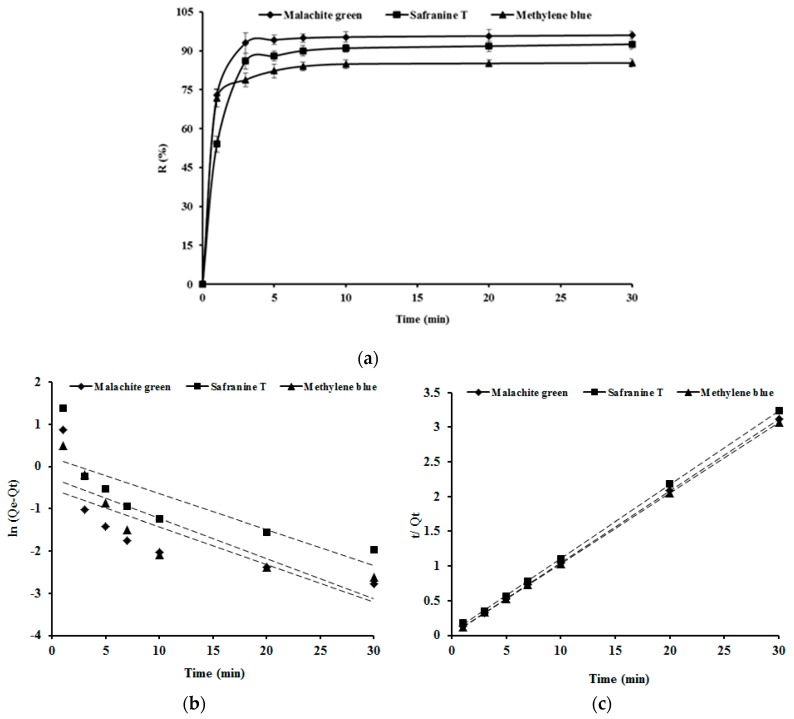
Adsorption removal efficiency of malachite green, safranine T and methylene blue by SSPE 1 g of SSFE with different times at 100 mg/L of initial dye concentration, pH 6.0 and 303 K of temperature. (**a**), Plots of first-order kinetics (**b**) and second-order kinetics (**c**). The solid lines are the experimental data, and the dashed lines are the linearized first-order and second-order kinetics fitted models.

**Table 1 ijerph-16-00679-t001:** Langmuir model coefficients.

Dye	*Qm* (mg/g)	*K_L_* (L/mg)	*R* ^2^
Malachite green	19.23	0.3071	0.986
Safranine T	42.02	0.0386	0.897
Methylene blue	18.45	0.5503	0.957

**Table 2 ijerph-16-00679-t002:** Freundlich model coefficients.

Dye	*n*	*K_F_* (mg/g) (L/mg)^1/n^	*R* ^2^	*n* *	*K_F_* * (mg/g) (L/mg)^1/n^	*R*^2^ *	Ref.
Malachite green	1.38	4.12	0.981	1.58	8.30	0.980	[25]
				2.18	26.27	0.980	[26]
Safranine T	1.11	1.55	0.995	1.28	15.50	0.990	[25]
Methylene blue	1.39	5.77	0.993	2.08	58.90	0.980	[25]
				2.04	19.31	0.983	[26]
				2.47	7.02	0.978	[27]
				1.05	1.13	0.999	[28]

* Results from related references.

**Table 3 ijerph-16-00679-t003:** Coefficients of adsorption thermodynamics.

Dye	$\Delta H\;(\mathrm{kJ}\;{\mathrm{mol}}^{-1})$	$\Delta S\;(J\;{\mathrm{mol}}^{-1}\;K^{-1})$	*R* ^2^
Malachite green	−10.35	−26.1	0.984
Safranine T	−23.99	−77.5	0.983
Methylene blue	−16.00	−42.6	0.981

**Table 4 ijerph-16-00679-t004:** Coefficients of the first and second order kinetic models.

Dye	First Order Kinetic Model		Second Order Kinetic Model		
	*k*_1_ (1/min)	*R* ^2^	*Qe* (mg/g)	*k*_2_ (g/mg)·(1/min)	*R* ^2^
Malachite green	0.0891	0.609	9.662	0.595	≈1
Safranine T	0.0849	0.652	9.389	0.259	≈1
Methylene blue	0.0948	0.725	9.823	0.503	≈1

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
