# Peer review of "Adsorption Thermodynamics and Dynamics of Three Typical Dyes onto Bio-adsorbent Spent Substrate of Pleurotus eryngii"

_ijerph, 2019, doi:10.3390/ijerph16050679_

Round 1

Reviewer 1 Report

Reviewers' comments:

Reviewer #1: The paper " Adsorption thermodynamics and dynamics of three typical dyes onto bio-adsorbent spent substrate of Pleurotus eryngii " is interesting but needs many adjustments to be accepted.

It is necessary to perform experiments on the physico-chemical characteristics of the adsorbent and the dye molecules.

It also needs a technical revision of the language and a better interpretation of the results obtained along with its justification. Here are some comments.

In pg 1, lines 26, 39, 44, 45, lacked references and lacked to better develop the studied subject

In pg 2, line 74, the authors wrote "Adsorption rate (%) ...", the correct term would not be percentage of adsorption (%)?

In pg 3, the authors only describe the results obtained but did not justify

In pg 4, lines 117-128, presented of organization, therefore this topic belongs to materials and methods

In general the results need to be reviewed and justified and the work needs to be better developed because it is very incomplete

Author Response

Response to Reviewer 1 Comments

Point 1: It is necessary to perform experiments on the physico-chemical characteristics of the adsorbent and the dye molecules.

Response 1: The chemical structures of the three dyes and FTIR spectra of SSPE were supplemented as Fig.1 and 3 in the revised manuscript (section 1 and 3.1).

Point 2: It also needs a technical revision of the language and a better interpretation of the results obtained along with its justification. Here are some comments.

In pg 1, lines 26, 39, 44, 45, lacked references and lacked to better develop the studied subject

In pg 2, line 74, the authors wrote "Adsorption rate (%) ...", the correct term would not be percentage of adsorption (%)?

In pg 3, the authors only describe the results obtained but did not justify

In pg 4, lines 117-128, presented of organization, therefore this topic belongs to materials and methods

In general the results need to be reviewed and justified and the work needs to be better developed because it is very incomplete.

Response 2: We have modified the manuscript based on the reviewer’s comments.

Reviewer 2 Report

The article titled "Adsorption thermodynamics and dynamics of three typical dyes onto bio-adsorbent spent substrate of Pleurotus eryngii" shows good results, but needs some adjustments to be accepted in this journal:

1. The introduction needs to be improved, also emphasizing the dye adsorption, dye types, structure, etc.

2. The adsorbent used needs to be characterized, and the functional groups capable of interacting with the adsorbate are quantified.

3. In line 97-99, the authors cite "... which indicated that the adsorption ...... and coordination reaction." How can a coordination reaction occur in the system used in the article?

4. A mechanism of interaction between the functional groups of adsorbate and adsorbents must be proposed.

5. In Figure 2a, concentrations less than 10 ppm (more diluted) should be used to try to obtain a curve coming out of the 0 point better, in order to obtain a more real adjustment to the models used.

6. In the thermodynamic parameters Gibbs free energy must be included.

7. Negative entropy means what?

8. The time (kinetic) isotherm must be repeated using a higher concentration, because of the way it was used, it is forcing a good fit to the pseudo second order model, and it may not be this way, since it is not encoded in the literature values of R2 = 1.

Author Response

Response to Reviewer 2 Comments

Point 1: The introduction needs to be improved, also emphasizing the dye adsorption, dye types, structure, etc.

Response 1: We have modified in the revised manuscript (section 1).

Point 2: The adsorbent used needs to be characterized, and the functional groups capable of interacting with the adsorbate are quantified

Response 2: The FTIR spectra of SSPE were added in the revised manuscript, and it was found that –OH and –COOH were the functional groups capable of interacting with the adsorbate (section 3.1).

Point 3:  In line 97-99, the authors cite "... which indicated that the adsorption ...... and coordination reaction." How can a coordination reaction occur in the system used in the article?

Response 3: I am sorry, it’s a clerical error. So “...through electrostatic attraction, complexation and coordination reaction” have been deleted in the revised manuscript (section 3.1).

Point 4: A mechanism of interaction between the functional groups of adsorbate and adsorbents must be proposed.

Response 4: The mechanism of interaction between the functional groups of adsorbate and adsorbents have been proposed in the revised manuscript (section 3.1).

Point 5:  In Figure 2a, concentrations less than 10 ppm (more diluted) should be used to try to obtain a curve coming out of the 0 point better, in order to obtain a more real adjustment to the models used.

Response 5: Yes, it is a very good suggestion.

Point 6: In the thermodynamic parameters Gibbs free energy must be included.

Response 6: Yes, the Gibbs free energy was added in the revised manuscript (section 3.3).

Point 7: Negative entropy means what?

Response 7: Negative entropy means that the degree of system chaos decreased. I have modified in the revised manuscript (section 3.3).

Point 8: The time (kinetic) isotherm must be repeated using a higher concentration, because of the way it was used, it is forcing a good fit to the pseudo second order model, and it may not be this way, since it is not encoded in the literature values of R2 = 1.

Response 8: Regression coefficient (R2) of the pseudo second order model closed to 1 (near 0.9999), so “1” was got by rounding in the manuscript. In order to follow the truth, “1” was replaced with “≈1” in the revised manuscript (Table 4).

Reviewer 3 Report

In the manuscript submitted by Jianguo Wu and co-authors, the authors investigated the adsorption characteristics of three typical dyes on bio-adsorbent (spent substrate of Pleurotus eryngii). The content of this paper is within the scope of the International Journal of Environmental Research and Public Health. Although the authors published a very similar research using Ganodorma lucidum in Bioresource Technology (266, pp134-138, 2018), the manuscript submitted to IJERPH also contains some novelty and inventiveness. Comments are listed below.

Comments

1.       Add chemical structures of the three dyes in a new figure. Because the adsorption characteristics of the dyes on SSPE probably depend on their chemical structures.

2.       Did the authors prepare control samples without SSPE in the batch bio-adsorption procedure? For example, they showed adsorption efficiencies of three dyes with different temperature in Fig.3. In these experiments, the authors should confirm the effects of heat decomposition of the dyes without SSPE. If the authors conducted control experiments, add detail in section 2.3. (section 2.3)

3.       The term “adsorption rate” is confusing because it sometimes means the adsorption speed. So, change the term to removal efficiency, for example. (section 2.5)

4.       Why pH affected the removal efficiency of safranine T? Add some discussion. (section 3.1)

5.       Add SSPE dose, initial dye concentrations, and contact time in the caption of Fig.1. (Fig. 1)

6.       Add SSPE dose, contact time, and pH in the caption of Fig.2-a. (Fig. 2)

7.       As shown in the Table 1, the Qm and Kf values of Safranine T were greatly different from those values of Malachite green and Mrthylene blue. Why? Add some discussion. (section 3.2)

8.       Add SSPE dose, initial dye concentrations,pH, and contact time in the caption of Fig.3-a. (Fig. 3)

9.       Add SSPE dose, initial dye concentrations, and pH in the caption of Fig.4-a. (Fig. 4)

10.     Among the three dyes, which dyes are most effectively removed by SSPE?

Author Response

Response to Reviewer 3 Comments

Point 1:  Add chemical structures of the three dyes in a new figure. Because the adsorption characteristics of the dyes on SSPE probably depend on their chemical structures.

Response 1: The chemical structures of the three dyes have been added as Fig.1 in the revised manuscript.

Point 2:  Did the authors prepare control samples without SSPE in the batch bio-adsorption procedure? For example, they showed adsorption efficiencies of three dyes with different temperature in Fig.3. In these experiments, the authors should confirm the effects of heat decomposition of the dyes without SSPE. If the authors conducted control experiments, add detail in section 2.3. (section 2.3)

Response 2: Yes, the control group was also conducted under the same batch bio-adsorption procedure of experimental group without SSPE, and C0 in Eq. (1) and (2) was the equilibrium liquid-phase concentrations of the dyes of control group, which was modified in the revised manuscript.

Point 3:   The term “adsorption rate” is confusing because it sometimes means the adsorption speed. So, change the term to removal efficiency, for example. (section 2.5).

Response 3: The term “removal efficiency by adsorption” was instead of the term “adsorption rate” in the revised manuscript.

Point 4: Why pH affected the removal efficiency of safranine T? Add some discussion. (section 3.1).

Response 4: Some discussion was added in the revised manuscript (section 3.1).

Point 5:  Add SSPE dose, initial dye concentrations, and contact time in the caption of Fig.1. (Fig. 1).

Response 5: SSPE dose, initial dye concentrations, temperature and contact time were added in the caption of Fig.2 showed in the revised manuscript.

Point 6: Add SSPE dose, contact time, and pH in the caption of Fig.2-a. (Fig. 2).

Response 6: SSPE dose, pH, temperature and contact time were added in the caption of Fig.4-a showed in the revised manuscript.

Point 7: As shown in the Table 1, the Qm and Kf values of Safranine T were greatly different from those values of Malachite green and Mrthylene blue. Why? Add some discussion. (section 3.2)

Response 7: Some discussion was added in the revised manuscript (section 3.2)

Point 8:  Add SSPE dose, initial dye concentrations,pH, and contact time in the caption of Fig.3-a. (Fig. 3).

Response 8: SSPE dose, initial dye concentrations, pH and contact time were added in the caption of Fig.5-a showed in the revised manuscript.

Point 9: Add SSPE dose, initial dye concentrations, and pH in the caption of Fig.4-a. (Fig. 4).

Response 9: SSPE dose, initial dye concentrations, pH and temperature were added in the caption of Fig.6-a showed in the revised manuscript.

Point 10: Among the three dyes, which dyes are most effectively removed by SSPE?

Response 10: Methylene blue was most effectively removed by SSPE.

Round 2

Reviewer 1 Report

The changes are in line with the suggestions. Thank you for your attention.

Reviewer 2 Report

Accept in present form.